# The Crystal Structure of the *Spodoptera litura* Chemosensory Protein CSP8

**DOI:** 10.3390/insects12070602

**Published:** 2021-07-01

**Authors:** Qian Jia, Hui Zeng, Jinbing Zhang, Shangfang Gao, Nan Xiao, Jing Tang, Xiaolin Dong, Wei Xie

**Affiliations:** 1MOE Key Laboratory of Gene Function and Regulation, State Key Laboratory for Biocontrol, School of Life Sciences, Sun Yat-Sen University, Guangzhou 510006, China; jiaq7@mail.sysu.edu.cn (Q.J.); zengh27@mail2.sysu.edu.cn (H.Z.); zhangjb66@mail2.sysu.edu.cn (J.Z.); gaoshf3@mail2.sysu.edu.cn (S.G.); xiaon8@mail2.sysu.edu.cn (N.X.); tangj63@mail2.sysu.edu.cn (J.T.); 2Forewarning and Management of Agricultural and Forestry Pests, Hubei Engineering Technology Center, Yangtze University, Jingzhou 434025, China

**Keywords:** crystal structure, CSP, rhodojaponin III, structure-function relationship, ligand-binding specificity

## Abstract

**Simple Summary:**

Worldwide, pest control involves extensive use of insecticides, which results in serious environmental pollution problems. On the other hand, insecticides can be recognized by proteins named CSPs in insects, which allow them to accurately respond to these environmental chemical signals for their survival, but the mechanism is poorly studied. Here, we report the crystal structure of the CSP8 protein from the tobacco cutworm *Spodoptera litura*, a major plant pest in Asia. We also studied its binding properties to compounds like rhodojaponin III, a non-volatile plant metabolite. Our studies showed that the protein binds to these molecules with different affinities and provided important insight into the molecular recognition mechanism of the sensory protein SlCSP8 and the CSP protein family in general.

**Abstract:**

*Spodoptera litura* F. is a generalist herbivore and one of the most important economic pests feeding on about 300 host plants in many Asian countries. Specific insect behaviors can be stimulated after recognizing chemicals in the external environment through conserved chemosensory proteins (CSPs) in chemoreceptive organs, which are critical components of the olfactory systems. To explore its structural basis for ligand-recognizing capability, we solved the 2.3 Å crystal structure of the apoprotein of *S. litura* CSP8 (SlCSP8). The SlCSP8 protein displays a conserved spherical shape with a negatively charged surface. Our binding assays showed that SlCSP8 bound several candidate ligands with differential affinities, with rhodojaponin III being the most tightly bound ligand. Our crystallographic and biochemical studies provide important insight into the molecular recognition mechanism of the sensory protein SlCSP8 and the CSP family in general, and they suggest that CSP8 is critical for insects to identify rhodojaponin III, which may aid in the CSP-based rational drug design in the future.

## 1. Introduction

Worldwide, pest control involves the extensive use of insecticides. The last three decades have witnessed great advantages and successes in food production, but the negative consequences of these chemicals used at large scales can never be underestimated. The extensive use of synthetic, broad-spectrum pesticides has the potential hazardous effects on the environment, human health, and so on [1].

The *CSP* gene was first discovered in 1994 and named factory-specific protein D (OS-D) [2]. Later, Angeli et al. found a class of proteins mediating chemical pheromone recognition in *Schistocerca gregaria*, which they renamed chemosensory proteins (CSPs) [3]. CSPs are present in chemoreceptive organs and can bind a range of aliphatic compounds, esters, and other long-chain compounds [2,4] to stimulate specific insect behaviors. CSPs consist of 100–115 amino acids with molecular weights of ~13 kDa. They are highly stable and are evolutionarily conserved except for their C-termini, suggesting that they play essential roles in insects’ lives. For example, the expression levels of some CSP proteins were significantly upregulated in the abdomen of *Bombyx mori* when it was treated with sublethal concentrations of avermectin [5]. Similar observations were obtained when *Terthron albovittata* was treated with thiamethoxam [5]. Furthermore, the expression levels of three CSP proteins (CSP4, CSP8, and CSP3) were upregulated in the head of *Plutella xylostella* in the presence of pyrethroids [6]. In addition, the expression of CSP2 in Lepidoptera was significantly increased in the presence of rhodojaponin III, a non-volatile secondary metabolite of plants [7]. Conversely, after blocking the CSP2 expression by RNAi, insects mistakenly laid eggs on rhodojaponin III-treated azalea, suggesting that CSP2 is implicated in identifying non-volatile metabolites in plants [8].

The approach to understanding the role of CSPs in the chemoreception process relies on a detailed knowledge of their three-dimensional structures and their binding modes, and possible conformational changes induced by the formation of complexes with their ligands [9]. The first structure of CSP (MbCSPA6) was reported from *Mamestra brassicae* (moth) [10,11]. Overall, MbCSPA6 is a globular protein and consists of six α-helices named helices A-F. A hydrophobic slit is formed in the center of the helices. Four conserved cysteines (Cys29-Cys36 and Cys55-Cys58) form two disulfide bonds and maintain a stable tertiary structure. Following the success on the apoprotein, the crystal structure of MbCSPA6 in complex with 12-bromododecanol was determined by Campanacci et al. in 2003 (PDB 1N8V) [12]. Significant conformational changes were found when 12-bromododecanol was bound. In order to accommodate the large ligand molecule, the binding cavity increased its volume and the position of each helix changed to various degrees. Of note, helix C displayed the most dramatic structural rearrangements and was pushed out by ~5 Å. Then the structures of *Bombyx mori* CSP1 and *Schistocerca gregaria* CSP4 were solved, respectively [9,13]. No further structures of CSPs were described and documented.

Although several structures of CSPs have been reported, our understanding of the ligand recognition and the binding modes by CSPs is still limited, given the large variety of CSPs. In addition, we are unclear about the molecular mechanism behind their physiological functions and therefore unable to carry out rational designs of pesticides due to a lack of structural information. Our previous study showed that rhodojaponin III might be a potential ligand of CSP [7,8]. To further study the structure–activity relationships of CSP, here we report the structure of CSP8 from *S. litura* (SlCSP8) and intended to provide evidence of its potential ligands. By gaining a better understanding of the olfactory system of *S. litura*, we expect to provide a new theoretical basis for the development and applications of novel insecticides.

## 2. Materials and Methods

### 2.1. Expression and Purification of SlCSP8

Total RNA of *S. litura* was isolated from twenty individual adults using Trizol according to the manufacturer’s specifications (Invitrogen, Carlsbad, CA, USA). First-strand cDNA was synthesized with the first-strand synthesis kit using Reverse transcriptase M-MLV (RNase H) (TaKaRa, Shiga, Japan). Briefly, 1 μg of total RNA, 1μL of Oligo (dT) primer (50 mM), and RNase-free deionized H_2_O were mixed, incubated at 70 °C for 10 min, and chilled on ice for 2 min immediately. Then, 0.5 μL of RTase M-MLV (RNase H), 2 μL of 5× M-MLV buffer, 0.5 μL of dNTP mixture (10 mM each), 0.25 μL RNase inhibitor were added to a final volume of 10 μL. The reaction mix was incubated at 42 °C for 60 min, 70 °C for 15 min, and cooled on ice. Then, the cDNA was stored at −20 °C. The gene encoding SlCSP8 was amplified by PCR from the cDNA of *Spodoptera litura* using the primers 5′-GCGGCAGCGGATCCGATGAAATTCGTACTAGTATTGTG-3′ and 5′-TTGCACTTCTCGAGTTCTGGGATGACGATGCCGTT-3′, respectively. The primer sequences contain the protective bases and restriction enzyme sites. After the double digestion by the restriction enzymes BamHI and XhoI, the PCR product was inserted into the pET-20b (+) (MerckMillipore) vector. The proteins expressed would possess a C-terminal 6× His tag. The D18-E128 fragment (SF) was subcloned in the same manner. The primers of the SF version are 5′-GCGGCAGCGGATCCGGATGAAAAGTACCCTAGCAAGTA-3′ and 5′-TTGCACTTCTCGAGTTCTGGGATGACGATGCCGTT-3′, respectively.

The plasmid was transformed into the *Escherichia coli* strain BL21 (DE3) cells for overexpression. When OD_600_ reached 0.6, expression of the target protein was induced with 0.2 mM IPTG, and the temperature was decreased to 25 °C. Cells were harvested after 16-h induction. The *E. coli* cells were pelleted by centrifugation at 3500× *g* for 20 min. Then, the cells were resuspended in a solution containing 30 mM Tris-HCl pH 8.0, 20% sucrose, and 1 mm EDTA. The precipitation was obtained by centrifugation at 23,500× *g* for 10 min at 4 °C, while the supernatant was removed. Repeat this step once. The precipitation was completely redissolved in 5 mM MgSO_4_ solution and slowly stirred on ice for 10 min. At this time, periplasmic proteins were released. After centrifugation at 23,500× *g* for 10 min, the supernatant was collected. All steps were carried out at 4 °C throughout. The protein was further subjected to ion exchange purification by a Q HP column (Cytiva) using a NaCl gradient. The purified protein was pooled, dialyzed in a buffer containing 20 mM Tris-HCl pH 8.0, 150 mM NaCl, and 1 mM DTT. The molecular weight of the protein was checked and confirmed by the MALDI-TOF mass spectrometry. The MALDI-TOF/TOF spectrometry was conducted on a UltrafleXtreme (Bruker, Karlsruhe, Germany) mass spectrometer operating in the positive ion mode, with sinapinic acid as the matrix. The protein sample was desalted prior to the analysis, and the signals between 10 and 15 kDa were scanned. The protein was stored after being flash-frozen in liquid nitrogen at −80 °C.

### 2.2. Crystallization, Data Collection, Phasing, and Data Processing

Initial crystallization screens were set up using the sitting-drop vapor diffusion method [14], and 12 mg/mL protein was mixed with an equal volume of the reservoir solution at 20 °C. Crystals of SlCSP8 were obtained in a condition of 2.5–3.0 M (NH_4_)_2_SO_4_, and 0.1 M NaOAc pH 5.5.

The diffraction data of SlCSP8 were collected using an Oxford Diffraction Xcalibur Nova diffractometer. The diffractometer was operated at 50 kV and 0.8 mA, with a rotation of 1 per frame at 120 °C. The data were recorded using a 65 mm Onyx CCD detector, and the exposure time was 90 s for each frame. The complete dataset was processed using CrysAlisPro (v.1.171.33.49; Oxford Diffraction, Abingdon, UK) and scaled using SCALA from the CCP4 suite. To solve the crystal structure, molecular replacement (MR) was first performed using Phaser [15] with the coordinates of the *Mamestra brassicae* CSPA6 (MbCSPA6) structure (PDB 1KX8) as the search model [11]. The protein model was further built manually according to the electron density map with COOT [16]. Multiple cycles of refinement alternating with model rebuilding were carried out by *PHENIX.*refine [17], and the final model was validated by Molprobity [18]. The structural figures were produced with PyMOL (http://www.pymol.org/) (accessed on 30 June 2021). All data collection and refinement statistics are presented in Table 1.

### 2.3. Fluorescence Assays

The assays were carried out with 3 μM protein in a buffer of 150 mM NaCl, 20 mM Tris-HCl pH 8.0. The concentrations of rhodojaponin III, bombykol, 12-bromododecanol, or avermectin (all purchased from Sigma ( St. Louis, MO, USA), purities ≥ 99%, HPLC) used were 0, 5, 10, 20, 40, 80, and 160 μM, respectively. The fluorescence spectra were conducted on an RF530R1PC fluorescence spectrophotometer (Shimadzu, Kyoto, Japan). The ligand interactions were monitored by the quenching of the protein fluorescence. The excitation was set at 280 nm, and the emission was at 290–450 nm, with a slit width of 1 nm.

## 3. Results

### 3.1. Expression and Purification of SlCSP8

The WT SlCSP8 protein was initially cloned into the pET-28a (+) plasmid (MerckMillipore) for overproduction in the cytoplasm of *E. coli*. However, it was not overexpressed, probably due to the failure to form the disulfide bonds in the reducing environment of the bacterial cytosol. We therefore adopted a similar strategy to what was employed by Campanacci, who utilized the pET-22b (+) vector to produce the MbCSPA6 protein [10]. Similarly, there is a pelB signal peptide (+) upstream of the gene of our interest in the pET-20b (+) vector (MerckMillipore), which translocates and localizes to the periplasm for expression of the fused target protein. Additionally, this peptide would be removed by the proteases when the fusion protein crosses the membrane. Therefore, the full-length gene and the D18-E128 fragment were subcloned into the pET-20b (+) vector. The N-terminus (M1-E17) is a putative signal peptide and was predicted to form a long helix, which is not observed in the CSP proteins whose structures have already been determined. The test expression showed that while the full-length protein was not expressed by the pET-20b (+) vector, the expression level of the truncation mutant in the same vector was high. After the initial purification by the Ni-NTA affinity purification step, 2 L of LB media generates ~10 mg protein with a purity of ~80%. A second step by anion exchange chromatography performed on a Q HP HiTrap column (Cytiva) removes most of the contaminant proteins and the resulting protein would be highly pure (95% purity). The MALDI-TOF mass spectrometry indicated that the average molecular weight of the target protein was consistent with the molecular mass of the D18-E128 fragment plus the C-terminal 6× His tag, within the MALDI instrumental errors. Consequently, we considered that the protein was successfully processed and that the sample was homogenous. The protein was concentrated to a high concentration of ~12 mg/mL and subsequently set up for crystallization.

### 3.2. Overall Structure of SlCSP8 and Structural Homologs

The structure of the SlCSP8 protein was determined at a high resolution of 2.3 Å. The protein was crystallized in the space group of *P*2_1_2_1_2_1_ and each asymmetric unit contains a single molecule (Figure 1a). The cloned fragment is completely visible, in addition to the C-terminal 6× His tag (the last histidine residue is missing). The overall structure is spherical, with the surface charges of the protein being mainly negative (Figure 1b). The N-terminal residues V17-D26 form a flexible loop, followed by six helices (N27–N122), which form the main body of the structure. The C-terminal tag adds an additional helix (α7) and extends to the solvent. The helices of the main body collapse onto each other and form a small central channel. α6 is parallel to the two planes formed by α2–α3 and α4-α5, respectively, but is perpendicular to α4. Two disulfide bonds are formed between C72 and C75, between C46 and C53, respectively. The refined model contains a total of 119 amino acids and 53 water molecules (Table 1). The topology of the domain is illustrated in Figure 1c.

Dali search for structurally similar proteins [19] resulted in several CSP orthologs from different organisms, and they are MbCSPA6 in the apo-form (PDBs 1KX8 and 1KX9), MbCSPA6 in complex with 12-bromododecanol (PDBs 1N8V and 1N8U), and SgCSP4 (PDB 2GVS), respectively [9,10,11]. These orthologs share 49–78% sequence identities with SlCSP8 (Figure 1d). The closest structural homolog apo-MbCSPA6 (78% sequence identity) could be aligned onto SlCSP8 with an RMSD of 0.96 Å over 100 Cα atoms, suggesting a high structural resemblance. The other orthologs could be superimposed onto SlCSP8 with RMSDs of 2.00–2.23 Å over more than 100 Cα atoms. By overlaying these structures, we found that the core domains of the sensory proteins are generally preserved among these proteins (Figure 2). Additionally, the multiple sequence alignment indicated that the primary sequences among different CSPs are highly similar. Of note, α1-α3 are more conserved than the other regions (Figure 1d).

### 3.3. The Binding of the Potential Ligands

After the structure determination of the apoprotein, we wondered if SlCSP8 binds several potential ligands: rhodojaponin III, bombykol, 12-bromododecanol, as well as avermection. In order to investigate the possible conformational changes upon the binding of the ligands, we conducted the binding assays on an RF530R1PC fluorescence spectrometer. The SlCSP8 protein generated its emission maximum at ~340 nm when it was excited at 280 nM wavelength. Upon the addition of 5 μM ligand of rhodojaponin III (1.7-fold of the protein concentration), the emission intensity dropped from 970 to 760 but the peak position remained the same. This >20% reduction in fluorescence suggested that the ligand was starting to bind the protein. The successive additions of the same ligand would continuously decrease the emission. At 160 μM ligand, i.e., when the ligand was in 50-fold excess, less than ~20% of the original peak intensity remained (Figure 3a). Consequently, we could conclude that SlCSP8 protein is capable of binding to rhodojaponin III strongly. Similarly, we performed the same experiments for the rest of the putative ligands. As indicated by Figure 3b–d, SlCSP8 would bind 12-bromododecanol slightly better than bombykol. By comparison, avermectin is the worst ligand. At a concentration of 160 μM, it barely decreased the emission of SlCSP8 and ~80% of the fluorescence was retained at the end of the titration.

With the knowledge that SlCSP8 is capable of binding various ligands with differential affinities, we tried the cocrystallization of SlCSP8 with each ligand. We obtained the crystals in the presence of these ligands in 10-fold excesses, collected the datasets, and solved the corresponding crystal structures. To our surprise, none of the solved structures contained the aforementioned ligands. We also tried soaking the crystals of the apoprotein with each ligand, but without any success either. One of the reasons could be attributed to the precipitation tendencies of the ligands dissolved in organic solvent in the crystallization drops, due to their poor solubilities in water. On the other hand, to find out a possible structural explanation, we compared the SlCSP8 structure with the CSP structures of *Mamestra brassicae* with and without the ligand 12-bromododecanol (PDBs 1KX9 and 1N8U, respectively) [11]. The structures are different from each other at their N-terminal loop regions, which point in different directions. However, the rest of the structures closely resemble one another. However, the binding of 12-bromododecanol brought conformational changes to MbCSPA6, mainly in the α2 helical region, in order to accommodate the linear ligand with a long hydrophobic tail. By employing the structural similarities between the two proteins, we generated the SlCSP8-bromododecanol complex model by superimposition. Here, SlCSP8 binds three ligand molecules. As shown by Figure 4a, all of the three ligand molecules are squeezed into a narrow space formed by the α3–α5 helices and pose steric clashes with the protein. However, close inspection of the complex model revealed that some of the clashes could be avoided through conformational changes as we observed in the MbCSPA6 complex structure. For example, although 12-bromododecanol molecules 1 and 2 (BDD1 and BDD2) would generate hindrances with the α2 helix of MbCSPA6, the clashes would be alleviated by the kink of this helix as well as the outward translational movement of α3 (Figure 4a). This is what indeed happed in the ligand-bound structure (PDB 1N8U) [12]. The residues involve Glu42-Gly54, including the completely conserved KELK motif. By contrast, the last 12-bromododecanol molecule (molecule 3) would butt against Tyr25 and Leu60, suggesting that a similar binding mode of the ligand at this site is unlikely. Therefore, due to the sequence differences, CSPs may display different preferences and affinities toward their ligands. Additionally, we compared the sizes, depths, and inner surface areas of the ligand-binding pockets of SlCSP8 and those of MbCSPA6, using the webserver Proteins Plus (https://proteins.plus) [20] (accessed on 30 June 2021). We found that SlCSP8 is significantly smaller in all three aspects than MbCSPA6 (Figure 4b), further suggesting differences in ligand identities and preferences between two CSP proteins.

## 4. Discussion

*Spodoptera litura* is a generalist herbivore [21]. Pesticides widely used for this pest have caused severe environmental pollution, including the enrichment of toxic molecules in the soil, underground water, and edible parts of the crops [22,23]. While the usage of pesticides enhances the production of agricultural foods, they also bring inestimable losses to our living environment. Therefore, insecticides specifically targeting key proteins in insects in recognizing these foreign substances could provide alternative biocontrol strategies. CSPs were originally found in the chemosensory system of insects, and it is believed to be involved in the chemo-recognition of different types of environmental chemicals [24]. Reports had suggested that CSPs can bind plant secondary metabolite molecules [25,26]. The most exciting properties of CSPs are their differential affinities to these hydrophobic molecules, which are conducive to the recognition and sense of these compounds by corresponding receptors. Although the idea of targeting chemical sensory proteins or odor-binding proteins was long proposed, no major breakthroughs have been achieved toward this goal due to a lack of data from structural, functional as well as physiological studies. A handful of CSP structures have been determined including several NMR structures, but currently little was known about these proteins, particularly on their ligand-binding properties. To date, only two CSP structures with bound ligands were reported (PDBs 1N8U and 1N8V, the ligand was 12-bromododecanol) [12]. Here, we conducted biochemical and structural studies on the CSP8 protein from *S. litura*, which is considered as a major pest to the agricultural industry and the overall national economy of China in the past decades.

We first tried the full-length protein as the subject of our studies. After the failure to express SlCSP8 in *E. coli* cytosol by the pET-28a (+) vector, the gene was subcloned into the pET-20b (+) vector, which was intended for periplasmic expression. To our disappointment, the full-length SlCSP8 protein was not over-expressed either. The N-terminus is a helical region, which was predicted to form a region of low complexity. In all of the solved CSP structures, none displays this helix. Therefore, we only worked on the truncated version D18-E128 for the follow-up structural and biochemical studies. The periplasmic expression of this construct produced the target protein in a large quantity. Here, we found that the lysis of the host bacterium by either osmotic shock or sonication did not make significant differences in terms of the yields or purities of SlCSP8. The partially purified protein was further purified through a second column to remove more indigenous proteins. The resulting protein was ~95% pure and generated rod crystals in a condition containing ammonium sulfate. We subsequently solved the crystal structure of SlCSP8*^D18-E128^*, which exhibited a spherical shape, very similar to the structure of MbCSPA6.

The folded CSP protein contains two intramolecular disulfide bonds, instead of the three found in odorant-binding proteins (OBPs). These disulfide bonds, conserved in CSPs, would generate loops of four and eight residues, respectively. Additionally, one of the cysteines is located on a loop, probably to strengthen the rigidity and reduce the flexibility of the protein. Other than the enhancement of thermostabilities of these sensory proteins, one would wonder whether they play any roles in other functions, such as the binding of hydrophobic molecules. A comparison of the structures of CSPs to those of OBPs revealed that there were barely any structural resemblances between the two types of proteins. However, both proteins can bind various hydrophobic molecules. An interesting problem: how would these proteins decide precisely which molecules they sense and yet work together?

We next conducted the binding experiments with several candidate ligand molecules. These molecules have been demonstrated to bind to different CSP proteins in the literature [8,12,27,28,29]. Our results showed that the SlCSP8 protein has the highest and lowest affinities to rhodojaponin III and avermectin, respectively, while it displays moderate affinities to 12-bromododecanol and bombykol, with the former being a better binder than the latter. Bombykol and 12-bromododecanol are long linear molecules with flexible hydrophobic tails, while rhodojaponin III and avermectin are fairly rigid molecules with multiple rings. Avermectin is a relatively large molecule with a molecular weight of 1732.1 Daltons. To bind this ligand with high affinity, SlCSP8 would either contain a large cavity on the surface to allow the avermectin to enter and bind, or it undergoes great conformational changes during the binding process. While it is not difficult to understand that avermectin has the lowest affinity to SlCSP8 due to its large size, it is interesting that a small protein like SlCSP8 would accommodate molecules with distinct chemical structures simultaneously. *Rhododendron molle* (B.) G. Don (Ericaceae) has long been used for insecticidal and medicinal purposes in China, and rhodojaponin III is the confirmed main component [30,31,32]. Rhodojaponin III is a grayanoid diterpene compound isolated from the flower of *Rhododendron molle*. Reports had documented that it has a high level of oviposition deterrent against more than 40 species of agricultural pests [33,34,35]. However, the mechanism of how these insects identify rhodojaponin III as an oviposition deterrent is yet poorly understood. We previously generated a docking model of the CSPSlit-rhodojaponin III complex [7]. We found that residues from the α2–α4 helices are the major contributors to the binding of this ligand. Coincidently, these regions are the most conserved regions among CSPs. SlCSP8 shares 48.8% sequence identity and 71.0% sequence similarity with that of CSPSlit [7]. The fluorescence assay performed in this study provides direct binding evidence of this compound by CSP and may offer an explanation to the reported observations.

Despite the potential of SlCSP8 to bind the ligands, our structural efforts to obtain their complexes either by cocrystallization or soaking were not successful, suggesting that the binding event may not easily occur in the crystallization environment. Because the ligands are generally hydrophobic, they had to be dissolved in organic solvents beforehand. When they were added to the crystallization drops, they tended to precipitate or crystalize due to their poor solubilities in water. On the other hand, we also compared the structures of apo-SlCSP8 with that of MbCSPA6 in the apo and 12-bromododecanol-bound forms; we found that at least some binding modes of the latter might not be allowed by SlCSP8, due to the local structural hindrances. However, this theory awaits the cocrystal structure of SlCSP8 in complex with candidate ligands. The fact that only two ligand-bound structures are available in the PDB database suggests the problematic nature of the cocrystallization. Kulmuni and Havukainen have studied the evolution of the CSP proteins, which they think are highly modifiable by their size, surface charge, and binding pocket. Additionally, they concluded that variations in the sizes of amino acid sidechains in the binding pocket would influence the ligand diversity [36]. Following the methodology of this study, we also investigated the sequence conservation of SlCSP8. Among the six variable ligand-binding residues listed by Kulmuni and co-worker, only A82 and A87 were conserved (PbarCSP1 numbering, Figure 1d). In addition, only K79 is conserved among the positively charged motif between helices 3–4 (K76, K78, K79 in PbarCSP10) while the glutamate residue is not conserved in the loop between helices 5–6 (E112 in PbarCSP10). The differences partially explained why there are so many CSPs for molecular recognition in insects and was also consistent with our modeling studies on the SlCSP8 complex.

CSPs play an important role in the insect’s sense of pesticides. Our study, despite the failure to reveal the ligand-binding patterns, characterized the apo-structure of SlCSP8 and carried out the binding assays with several important ligands. Follow-up studies would focus more on the contact points for ligand binding. Moreover, we found that SlCSP8 displays many commonalities with members in the CSP family, which suggested that the results derived from this study could also be applied to other members in general. Our studies contribute to a better understanding of the structure–activity relationship of this unique type of proteins.

## Figures and Tables

**Figure 1 insects-12-00602-f001:**
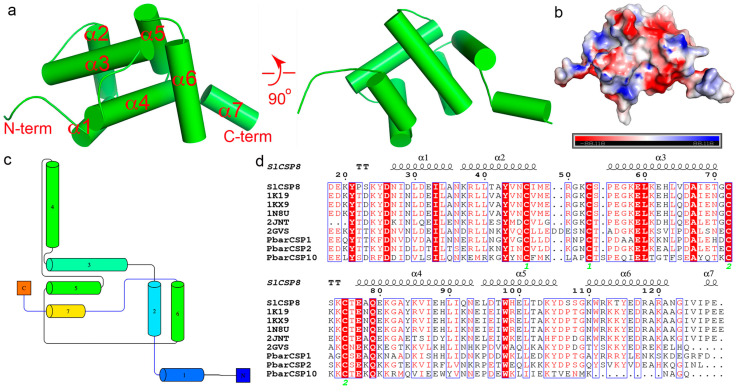
The structure of SlCSP8 and its sequence alignment with related CSP proteins. (**a**) The overall structure of SlCSP8. The structure was shown in cartoon in two orthogonal views. The N- and C-termini were indicated and the helices were labeled. The helices were shown by the cylinders. (**b**) The surface charge representation of SlCSP8, which was held in the same viewpoint as that in Figure 1a. The positively charged patch was colored blue while the negatively charged patch was colored red. (**c**) The topology of SlCSP8, as generated by the program Pro-origama (http://munk.csse.unimelb.edu.au/pro-origami/index.shtml) (accessed on 30 June 2021). The colors change from cold to warm as the sequence proceeds from the N- to the C-terminus. (**d**) Multiple sequence alignment of SlCSP8 and other CSPs with published structures. The secondary structure elements were labeled above the sequences. The green numbers “1” and “2” indicated the cysteine residues forming the two disulfide bonds. 1KX9, 1N8U and 1K19: *Mamestra brassicae* CSPA6; 2JNT: *Bombyx mori* CSP1; 2GVS: *Schistocerca gregaria* CSP4; PbarCSP1/2/10: *Pogonomyrmex barbatus* CSP1/2/10.

**Figure 2 insects-12-00602-f002:**
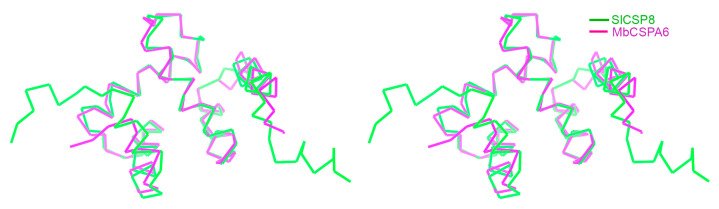
Structure comparison. The structural superimposition of the backbone structure of SlCSP8 (PDB 7E8L, colored green) onto that of MbCSPA6 (PDB 1KX9, magenta). The figure was shown in cross-eyed stereo view.

**Figure 3 insects-12-00602-f003:**
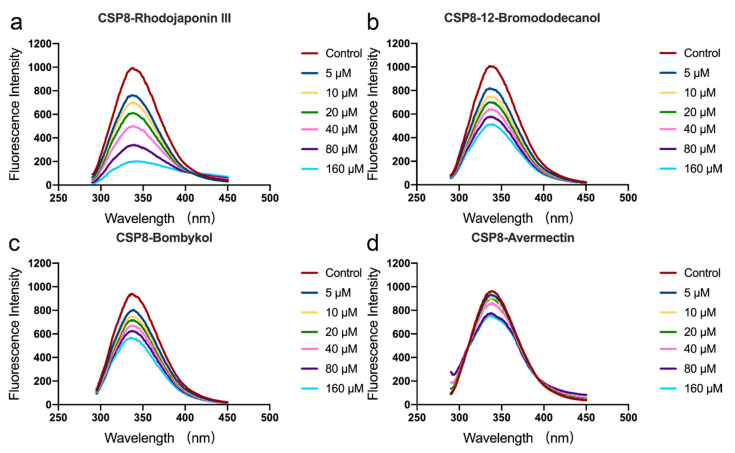
The fluorescence quenching of SlCSP8 by the titrations of various ligands (**a**–**d**). The emission reduction of the fluorescence at 340 nm by the addition of rhodojaponin III (**a**), 12-bromododecanol (**b**), bombykol (**c**), and avermectin (**d**). The vertical axis indicated the fluorescence intensities while the horizontal axis indicated the wavelengths. Control: The SlCSP8 sample with only buffers added. Seven concentrations of the ligands were employed and their influences were indicated by the curves of different colors.

**Figure 4 insects-12-00602-f004:**
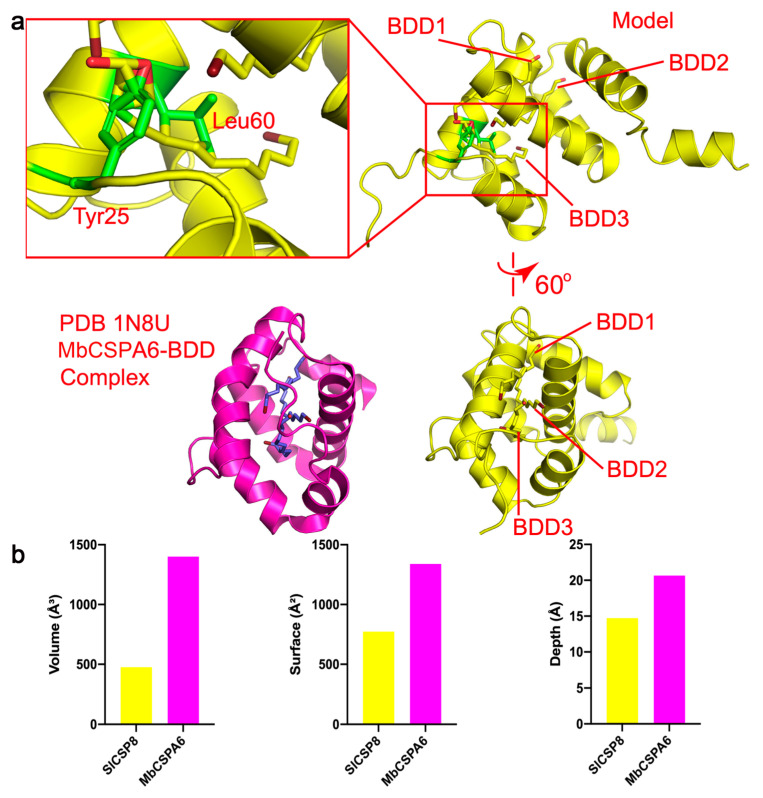
The properties of the ligand-binding pocket. (**a**) The SlCSP8-12-bromododecanol (BDD) model and their possible hindrances. The SlCSP8 protein was shown in two views, related by a rotation of 60 degrees around the *Y*-axis. The upper-left panel indicated the clashes between BDD3 and Tyr25 and Leu60. The lower-left showed the cocrystal structure of MbCSPA6 bound with the three BDD molecules (PDB 1N8U), which avoided the clashes through large conformational changes in SlCSP8. (**b**) The comparison of the volume sizes, depths, and inner surface areas of the ligand-binding pockets of SlCSP8 and those of MbCSPA6.

**Table 1 insects-12-00602-t001:** Data collection and refinement statistics.

Data Set	CSP8
**PDB ID**	7E8L
Resolution (Å)	23.70–2.30 (2.42–2.30) ^a^
Space group	*P*2_1_2_1_2_1_
Cell dimension (Å)	
*a*, *b*, *c* (Å)	38.40, 50.05, 60.24
*α*, *β*, *γ* (°)	90, 90, 90
R_merge_ ^b^	0.062 (0.196)
Redundancy	4.1 (3.5)
Completeness (%)	99.7 (99.0)
*CC* _1/2_	0.998 (0.958)
*I/* *σ_(I)_*	17.9 (6.4)
Refinement	
Resolution range (Å)	23.71–2.30 (2.36–2.30)
No. reflections	5212
R_work_ ^c^/R_free_ ^d^ (%)	20.83/26.07
No. atoms	
Protein	935
Water	53
B-factor (Å^2^)	
Protein	29.693
Water	31.349
R.m.s deviations	
Bonds (Å)	0.008
Angles (°)	1.309
Ramachandran favored (%)	94.87
Outliers (%)	0.2

^a^ Values in parentheses are for the highest-resolution shell. ^b^ R_merge_ = Σ |(*I* − < *I* >)|/*σ**_(I)_*, where *I* is the observed intensity. ^c^ R_work_ = Σ*_hkl_* ||*F*o| − |*F*c||/Σ*_hkl_*|*F*o|, calculated from working data set. ^d^ R_free_ is calculated from 5.0% of data randomly chosen and not included in refinement.

## Data Availability

Atomic coordinates and structure factors for the reported crystal structure has been deposited with the Protein Data bank under accession number 7E8L.

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
