# Peer review of "The Crystal Structure of the Spodoptera litura Chemosensory Protein CSP8"

_insects, 2021, doi:10.3390/insects12070602_

Round 1

Reviewer 1 Report

Comments:

This is a nice, reasonably well-executed study on the Spodoptera litura chemosensory protein CSP8. The authors report its crystal structure of the apo-protein, and investigate the binding properties to several potential ligands (rhodojaponin III, bombykol, 12-bromododecanol and avermectin) as well as the possible conformational changes upon the binding of the ligands. Through the structural comparison to that of MbCSPA6 in the apo- or 12-bromo-dodecanol bound forms, they found that one of the three bromo-dodecanol molecules would be in close proximity to Tyr25 and Leu60, suggesting that CSP8 would adopt different binding mode and display different preferences toward the ligands.

Insect olfaction is the basis of insect behavior. The study by Jia et al. promotes current understanding on insect behavior regulators and may provide an alternative strategy to control pests like Spodoptera litura.

Minor issues: 

  1. In the text, line 41 on pg2, the sentence“The CSP gene was first in 1994 and named factory specific protein D (OS-D)should be changed to The CSP gene was first discovered in 1994 and named factory specific protein D (OS-D)”.
  2. Line 61 onpg2, delete “analyzed”.
  3. Line 88-89 on pg2-3, there were extra spaces in the primer sequences.
  4. Line 104 on pg3, the “26-36% 2.5-3 M (NH4)2SO4” description was confusing.
  5. Line 161 on pg5,the surface charge representation in figure 1a should be an independent panel.
  6. Line 198 on pg6, 12-bromododecanol and 12-bromo-dodecanol were used interchangeably in some places. Please be consistent.  
  7. Line 292 on pg9, this was the first time that “OBPs” appears in this paper. The author should write down the full name.
  8. The protocol for osmotic shock should be more precise.

Reviewer 2 Report

The manuscript by Zhang et al describes generation and analysis of the Spodoptera litura chemosensory protein 8 (CSP8) crystal structure. The authors compare the conformation with that of previously determined CSP structures and assess potential ligand binding via a reduction in fluorescence emission.  

Although the aim of determining the CSP8 crystal structure expands the currently limited set of available structures, the manuscript does not sufficiently address why/how the particular CSP8 structure would provide new insights into the CSP binding pocket (ie why was CSP8 chosen for structure determination?). It is also unclear how the CSP8 in the current study compares with the CSP reported previously in Zhang et al 2013. Are they the same proteins? If so, how does the homology-based structure and the predicted ligand contact points compare with the solved structure? The same questions could be asked if the two proteins are in fact not the same. In addition, given the availability of the new structure, it is somewhat surprising that no predictions were made re ligand binding and/or ligand contact points that could be tested either in the current study or in follow up studies. Lastly, in lines 247-248 the authors speculate that differences in primary structure may impact ligand binding affinities. Kulmuni and Havukainen (2013; https://doi.org/10.1371/journal.pone.0063688) predicted that variations in the sizes of amino acid sidechains that compose the binding pocket impacted ligand diversity. Does the CSP8 structure support those predictions?

Other comments

1) The manuscript would greatly benefit from more thorough English editing.

2) More details should be included in the Methods section, which could involve moving significant portions of Results 3.1 to the methods.

- what is the template source of the cDNA used in line 87? How was the cDNA generated?

- are restriction enzyme sites included in the primers? If so, that should be indicated.

- what is the source of the pet-20b(+) vector?

- was the His tag cleaved? If not, why not?

- what was the volume of culture used?

- provide a reference for the sitting-drop vapor diffusion method

- define what the reservoir solution consists of.

- provide the source/supplier for the diffractometer.

- provide a reference for PHENIX

- provide sources and purity for the compounds listed on line 121.

- line 141 lists a D18-E128 fragment but this was not detailed in the Methods section

- line 150 indicates that MALDI-MS based analyses were conducted but this not detailed in the Methods section

3) Line 143 seems to indicate that previously determined CSP structures lack a signal peptide. Please clarify. Do the authors mean that the structures were generated with the signal peptide removed or that the proteins themselves lack a signal peptide? The previous lines indicate that the pelB signal peptide incorporated into the recombinant protein from the expression vector is proteolytically removed when translocated across the membrane. Perhaps clarify why a similar mechanism cannot (or will not) likewise remove the endogenous signal peptide.

4) In Results section 3.3, it is unclear why the compounds assayed were selected. Is there a biological reason for their inclusion? Also, it is unclear why a competitive binding assay was not used to assess ligand specificity.

5) Line 235 indicates that a SlCSP8-bromododecanol complex model was generated, but it is unclear if this model was only generated in silico or not. If so, this was not detailed in the Methods section. It is also unclear why additional compounds were not similarly docked using the CSP8 structure.

6) Comments re the Discussion

- lines 295-296 - provide relevant reference(s)

- line 310 – could variations in pH drive conformational changes similar to those observed with OBPs (see Leite et al 2009; https://doi.org/10.1371/journal.pone.0008006 and related studies)?

Reviewer 3 Report

Authors characterized the crystal structure and ligand binding of another lepidopteran CSP. This not novel, but may be of general interest, because Spodoptera litura is a serious global pest insect and knowledge on its CSPs may help to develop and apply ecofriendly pesticides. Unfortunately, authors were not able to over-express the full length SlCSP and had to use a truncated protein for their experiments. Nevertheless, I recommend to accept the manuscript after revision.

Specific comments:

  • Simple Summary has to be revised (spaces between words, grammatical errors etc.)
  • Line 41: was first described in …
  • Line 53 and others: either use “in Lepidoptera” or “in lepidopterans”
  • Line 61: delete “analyzed”
  • Line 82 and others: Spodoptera is correct
  • Line 100: samples were stored at minus (!) 80 degrees
  • Legend to Fig. 1: acronyms used for the published CSP should be explained here
  • Line 263: believed
  • Line 273: were reported
  • References have to be written according to Authors’ Instructions of MDPI (species names in italics; correct acronyms for journal titles etc.)

Author Response

Comment 1: Authors characterized the crystal structure and ligand binding of another lepidopteran CSP. This not novel, but may be of general interest, because Spodoptera litura is a serious global pest insect and knowledge on its CSPs may help to develop and apply ecofriendly pesticides. Unfortunately, authors were not able to over-express the full length SlCSP and had to use a truncated protein for their experiments. Nevertheless, I recommend to accept the manuscript after revision.

Response 1: We thank the reviewer for the summary of our work and we think that our study provides insight into the understanding of CSPs.

Comment 2: Simple Summary has to be revised (spaces between words, grammatical errors etc.)

Response 2: Sorry for our carelessness here. We have revised this section in the new version.

Comment 3: Line 41: was first described in …

Response 3: We have rewritten the sentence as The CSP gene was first discovered in 1994 and named factory specific protein D (OS-D).

Comment 4: Line 53 and others: either use “in Lepidoptera” or “in lepidopterans”

Response 4: We thank the reviewer for the suggestion. We have changed it to “in Lepidoptera”.

Comment 5: Line 61: delete “analyzed”

Response 5: Sorry for our mistake here. “analyzed” was deleted in the new version.

Comment 6: Line 82 and others: Spodoptera is correct

Response 6: Sorry for our carelessness here. We have changed it in the text.

Comment 7: Line 100: samples were stored at minus (!) 80 degrees

Response 7: We thank the reviewer for the advice. We have changed it to “-80℃”.

Comment 8: Legend to Fig. 1: acronyms used for the published CSP should be explained here

Response 8: We thank the reviewer for the advice. In the revised version, we have explained the published CSP in the legend of Figure 1, “1KX9, 1N8U and 1K19: Mamestra brassicae CSPA6; 2JNT: Bombyx mori CSP1; 2GVS: Schistocerca gregaria CSP4.”

Comment 9: Line 263: believed

Response 9: We thank the reviewer for the comment and we changed it in the revised text.

Comment 10: Line 273: were reported

Response 10: We thank the reviewer for the comment and we changed it in the revised text.

Comment 11: References have to be written according to Authors’ Instructions of MDPI (species names in italics; correct acronyms for journal titles etc.)

Response 11: In the revised version, we have rewritten the references.

Round 2

Reviewer 2 Report

Although the authors have addressed some of the comments, more extensive elaboration within the manuscript itself is desired, in particular in terms of how the current study may expand on previous studies and inform/guide future studies.  As stated previously, the manuscript does not sufficiently address why/how the study expands, previous in silico based structural analysis of CSPs from this species. This could be addressed by incorporating author responses into the manuscript and providing more direct comparisons that either support or refute findings in those studies. Are there shared contact points (potential conservation of binding)? Do the authors see obvious differences in the residues that comprise the proposed binding pocket that could provide insights re differential ligand binding? Again, it is surprising that no predictions were made re ligand binding and/or ligand contact points that could be tested either in the current study or in follow up studies.

Other comments –

1) English editing can be further improved.

2) Was a different primer used to construct the D18-E128 fragment? If so, it should be listed. Line 90 indicates that 1 mg of total RNA was used to generate cDNA. Is this correct or did the authors mean that 1 ug total RNA was used?

3) Line 116 indicates that the recombinant protein was confirmed by MALDI-TOF mass spectrometry. However, no details have been provided. What instrument was used? What matrix was used? What ionization method was used? If a defined protocol was used, then an appropriate paper should be referenced. What is the symbol at the end of the sentence in line 117?
